# Parent Stress as a Consideration in Childhood Obesity Prevention: Results from the Guelph Family Health Study, a Pilot Randomized Controlled Trial

**DOI:** 10.3390/nu12061835

**Published:** 2020-06-19

**Authors:** Valerie Hruska, Gerarda Darlington, Jess Haines, David W. L. Ma

**Affiliations:** 1Department of Human Health and Nutritional Sciences, University of Guelph, Guelph, ON N1G2W1, Canada; vhruska@uoguelph.ca; 2Department of Mathematics and Statistics, University of Guelph, Guelph, ON N1G2W1, Canada; gdarling@uoguelph.ca; 3Department of Family Relations and Applied Nutrition, University of Guelph, Guelph, ON N1G2W1, Canada; jhaines@uoguelph.ca

**Keywords:** stress, mental health, family, health behavior, childhood obesity, health intervention

## Abstract

Parents’ stress is independently associated with increased child adiposity, but parents’ stress may also interfere with childhood obesity prevention programs. The disruptions to the family dynamic caused by participating in a behaviour change intervention may exacerbate parent stress and undermine overall intervention efficacy. This study explored how family stress levels were impacted by participation in a home-based obesity prevention intervention. Data were collected from 77 families (56 fathers, 77 mothers) participating in the Guelph Family Health Study (GFHS), a pilot randomized control trial of a home-based obesity prevention intervention. Four measures of stress were investigated: general life stress, parenting distress, depressive symptoms, and household chaos. Multiple linear regression was used to compare the level of stress between the intervention and control groups at post-intervention and 1-year follow-up, adjusted for baseline stress. Analyses for mothers and fathers were stratified, except for household chaos which was measured at the family level. Results indicate no significant differences between intervention and control groups for any stress measure at any time point, indicating a neutral effect of the GFHS intervention on family stress. Future work should investigate the components of family-based intervention protocols that make participation minimally burdensome and consider embedding specific stress-reduction messaging to promote family health and wellbeing.

## 1. Introduction

Childhood overweight and obesity are associated with several health concerns such as increased risk of chronic illnesses like cardiovascular disease, type 2 diabetes, cancer, and reduced overall lifespan, as well as increased risk of being bullied and developing disordered eating habits due to societal bias against those in larger bodies [1,2,3,4]. While there is a well-recognized genetic predisposition to body composition, the main focus of childhood obesity prevention has been on health behaviours such as dietary patterns, physical activity, sedentary or screen-based time, and sleep quality. There appears to be a critical window of development in early childhood where lifelong health behaviour patterns are largely established [5,6]. This presents an especially advantageous target for programs to focus on prevention in early life to maximize the preventative benefit of healthful behavioural patterns. Parental involvement has repeatedly been demonstrated to play a key role in the success of childhood obesity prevention programs [7,8,9,10]. These family-based behaviour change interventions typically focus on changing parenting practices and/or family behaviours such as eating meals as a family or group physical activities. However, parents engaged in a home-based childhood obesity prevention program manage several roles; they are participants making changes to their own behaviours plus being the taskmaster for their child’s compliance, as well as the many other roles that they serve outside of the intervention context. The competing demands on parents’ time and resources are numerous and dynamic, making it especially complex to effectively engage them in childhood obesity prevention programs.

Parents’ stress may be an additional key consideration for family-based childhood obesity prevention programs for two key reasons. First, past research has established cross-sectional associations between parent stress or household dysfunction and several child health outcomes, including behaviours such as increased screen viewing [11], fast food consumption [12] as well as overall child weight status [12,13,14,15]. The second consideration is that parents who are overwhelmed may have difficulties adhering to an obesity prevention program, thus undermining the program’s efficacy. It is well-understood that family routines are an important contributor to family well-being and positively influence children’s development [16,17,18,19], but participation in a family-based childhood obesity prevention program is likely to impose substantial changes in the families’ typical routines and activities. This perturbation of existing habits, even if intended for healthful changes, may inadvertently disrupt balances within the home. Alternatively, it is possible that promoting new behaviours as part of healthful routines could help families to establish more order and regularity within the home, thus decreasing overall family stress. The impact of health promotion programs on parents’ wellbeing has not been widely explored.

In addition, dominant expectations of parenting place much more responsibility on mothers than fathers for active management of children’s health and health behaviours [20,21]. Studies in Canada, the US, and Europe consistently demonstrate that, despite men’s increasing involvement, women take on the bulk of responsibility for house and family work, including assuming responsibility for the health and well-being of family members, organizing their children’s lives, and planning and preparing meals [20,21]. Thus, family-based health interventions may inadvertently reinforce the gendered division of labour and could result in an enhanced level of stress among mothers as compared to fathers. Additionally, perceptions and consequences of stress have repeatedly been demonstrated to differ between males and females [22,23,24,25,26], thus making gender an important consideration when exploring how participation in a family-based intervention may influence family stress.

The purpose of this study was to investigate the longitudinal changes in parents’ perceived general life stress, parenting distress, depressive symptoms, and household chaos as a function of participation in a family-based health promotion intervention program among a cohort of Canadian mothers and fathers of young children. This study also examined whether these changes in family stress were moderated by parent gender.

## 2. Materials and Methods

### 2.1. Study Participants

This study used the Pilot phase 1 and 2 studies of the Guelph Family Health Study (GFHS), a pilot randomized control trial of a home-based obesity prevention intervention (clinical trials registration number NCT02223234, University of Guelph Research Ethics Board REB14AP008). The primary aim of the pilot studies was to test the feasibility of the intervention and assessment protocols. Detailed procedures of the pilot are published elsewhere [27] and briefly summarized below. Participants were recruited using posters and rack cards displayed at local family health team and early childhood education centres as well as posts to these agencies’ social media accounts. To be eligible to participate, families had to have at least one child between the ages of 18 months to 5 years of age, live in Wellington County, Ontario, Canada, with no plans to move in the following year, and have at least one parent able to respond to surveys in English. 

Data for these analyses were collected at baseline, 6-months (post-intervention) and 18-months (1-year post-intervention). Participating families received grocery gift cards as compensation at each time point of assessment. 

### 2.2. Exclusions and Losses to Follow-Up

As shown in Figure 1, 151 parent participants from 86 families met eligibility criteria and were enrolled in the study, though three families (three mothers, one father) later declined to participate before completing baseline assessment. The remaining 83 families (147 parents; 83 mothers, 64 fathers) were randomized to the three treatment groups: two home visits with a health educator (2HV), four home visits with a health educator (4HV), and a minimal-attention control, the protocols of which are explained further below. One family (one mother) randomized to the 4 HV group later declined to receive the intervention and was eventually lost to follow-up. The remaining 82 families (146 parents) completed all components of the intervention program, though five families (five mothers, six fathers) were later lost to follow-up, resulting in a 92.8% retention rate of the GFHS Pilot 1 and 2 cohorts at 1-year post-intervention. No harms of the intervention were detected. 

In addition to the 11 participants who were lost to follow-up, two fathers did not complete baseline stress measures and were therefore excluded from this analytic sample. Thus, a final analytic sample of 133 parent participants (77 mothers, 56 fathers) from 77 families was used for these analyses. 

### 2.3. GFHS Intervention

The GFHS was designed as a home-based childhood obesity prevention program, informed by the Family Systems [28] and Self Determination [29] theories. The program used motivational interviewing, a collaborative and client-centred counselling technique that increases the likelihood of successful behaviour change by providing families with a sense of autonomy, confidence, and support with respect to health goals that the families set for themselves. Suggested goals in the GFHS included increasing fruit and vegetable intake, replacing sugar-sweetened beverages with water, reducing screen time, establishing a bedtime routine to promote adequate sleep, encouraging physical activity, or another goal of the family’s own creation. The intervention program was delivered by a health educator, a registered dietitian trained in motivational interviewing, who worked with the families to develop personalized and self-directed health goals and provided support throughout the 6-month intervention period. These sessions were held in the family’s home and typically were an hour in duration. Complementary to the home visits were a series of emails and mailed materials tailored to the family’s goals, such as colourful plates to encourage more family meals or children’s books to encourage regular sleep routines. Full details of the intervention protocol have been published previously [27].

All participants completed the baseline assessment, including a series of surveys and health visits at the University of Guelph, where measurements such as height, weight, blood pressure, and body composition were taken by trained research assistants. After baseline assessment families were randomized by the study coordinator into one of three parallel groups (in Pilot 1) or into one of two parallel groups (in Pilot 2) using a pseudo-random number generator. The three groups in Pilot 1 consisted of a minimal-attention control group (general health advice through monthly emails, such as current Canadian physical activity guidelines), a two home visit intervention group (home visits with a health educator, weekly emails, and monthly mailed incentives), and a four home visit intervention group (differing only in number of visits from the two home visit group). In Pilot 2, families were randomized to control or four home visits based on early feedback from Pilot 1 participants that two home visits were not preferred. Baseline data were collected between December 2014 and November 2016 at the University of Guelph, Ontario, Canada; follow-up data collection was completed by November 2018.

### 2.4. Stress Measures

Four different types of stress (general life stress, parenting distress, parental depression, and household chaos) were assessed via paper (*n* = 152) or online (*n* = 238) surveys. Data collection was conducted at baseline, then repeated post-intervention (6 months from baseline) and at 1-year post-intervention (18 months from baseline). 

General life stress was examined with the question “Using a scale from 1 to 10, where 1 means ‘no stress’ and 10 means ‘an extreme amount of stress’, how much stress would you say you have experienced in the last year?” [12]. 

Levels of stress specific to the role of being a parent were examined using the 12-item Parent Distress subscale of the Parenting Stress Index (PSI) [30]. Participants were asked to respond on a 5-point Likert scale from 1 (strongly disagree) to 5 (strongly agree) to items such as “I often have the feeling that I cannot handle things very well”, “I feel trapped by my responsibilities as a parent”, and “Having a child has caused more problems than I expected in my relationship with my spouse (or male/female friend)”. For parents who completed the paper version of the surveys, the response options were on a 4-point Likert scale (i.e., the neither disagree nor agree option was not included). This discrepancy in the response options between the paper and online surveys was managed by recoding the paper survey response options as 1 = strongly disagree, 2 = disagree, 4 = agree, and 5 = strongly agree. Analyses with the paper and online survey data together showed similar results to when only the online survey data were used; thus, results for the paper and online survey were combined for these analyses. A total score out of 60 was calculated by summing the responses; higher scores indicate greater parental distress. Standardized Cronbach’s alpha for mothers in this sample at baseline, was 0.86; for fathers, 0.78. The PSI has been validated for use among both mothers [30] and fathers [31] of young children. 

Parental depressive symptoms were assessed with the Andresen short form of the Centre for Epidemiological Studies Depression Scale (CES-D) [32]. Sample items include “My sleep was restless”, “Everything I did was an effort”, and “I felt fearful”, and were scored as 0 (less than one day last week), 1 (1–2 days), 2 (3–4 days), or 3 (5–7 days). A total score out of 30 was calculated by summing the responses; higher scores indicate greater depressive symptoms. Standardized Cronbach’s alpha for mothers in this sample at baseline was 0.87; for fathers, 0.80.

Household dynamic and chaos were examined using the 15-item Confusion, Hubbub, and Order Scale (CHAOS) [33]. This scale conceptualizes noisiness, disorganization, and confusion within the home environment. Participants responded to items such as “We almost always seemed to be rushed” or “It’s a real zoo in our home” on a 4-point Likert scale from 1 (very much like your own home) to 4 (not at all like your own home). The CHAOS survey was asked only of Parent 1 in this sample (the first parent to sign up for the study, of whom 76% were female), and this was used as a family wide measure. Standardized Cronbach’s alpha for this scale at baseline was 0.88. 

### 2.5. Statistical Methods

In intent-to-treat complete case analyses, we used multiple linear regression models to examine differences between the study groups (control, 2HV, and 4HV) for post-intervention and for 1-year follow-up stress measures after controlling for baseline. Results for the 2HV and 4HV groups were not substantively different (see Table A1), thus, we present results with the two intervention groups combined. General stress, parenting distress, and depressive symptoms were analysed for each participant; household chaos was considered to be a shared variable among family measures and was analysed at the household-level. Data from males and females were analysed separately to account for potential gender-based differences in stress perception [22,23,24,25,26] and to better compare these results to the predominantly mother-focused parenting research in the field [34]. Household chaos was examined as one observation per family, regardless of the gender of the parent who reported it. No demographic covariates were included in the model. The use of a randomized design would mean that any difference in demographic characteristics across study groups would be due to chance. Statistical analyses were performed using SAS University Edition Version 3.6 [35]. A *p*-value of < 0.05 was considered statistically significant for all analyses.

## 3. Results

### 3.1. Descriptive Data

As shown in Table 1, this analytic sample contained 56 fathers (42%) and 77 mothers (58%). The average age of participants at baseline was 35 years. Over 80% of participants identified as white and over 40% had received postgraduate education. Of the 77 participating families, approximately 85% had parents who were married, nearly 80% contained two or more children, and 45% had an annual household income of $100,000 or more. Baseline characteristics (Table 1) and levels of stress (Table 2) were similar among the intervention and control groups. 

### 3.2. Mean Stress Levels

As shown in Table 2, mothers and fathers reported moderate levels of stress on all measures at all time points and across all treatment groups. Across the three timepoints, mothers’ general stress means ranged from 6.0 to 6.6 out of a maximum score of 10. Fathers’ general stress scores ranged from 5.9 to 6.8. Mothers’ parenting distress mean scores ranged from 26.8 to 30.9 out of a maximum score of 60, which ranks between the 59th and 68th percentiles of the PSI scoring reference [30]. Fathers’ parenting distress scores ranged from 27.5 to 29.0, which falls within the 62nd and 64th percentiles. Mothers’ depressive symptoms scores ranged from 6.0 to 6.8; fathers’ scores ranged from 6.1 to 7.9. While these CES-D means may seem low in relation to the maximum score of 30 points, they should be interpreted as moderate given that a CES-D score of 10 or greater indicates significant depressive symptomology consistent with clinical diagnosis [32]. Household chaos means ranged from 30.3 to 33.0 out of a maximum score of 60 points. 

### 3.3. Post-Intervention

No intervention effect was observed for any of the stress measures among mothers or fathers at post-intervention after controlling for baseline measures. Among mothers randomized to the intervention, there was a non-significant difference of −0.60 (95% CI: −1.47, 0.27) compared to control, after adjustment for baseline. Among fathers, there was a non-significant difference of 0.56 (95% CI: −0.43, 1.56) in the intervention compared to control, after adjustment for baseline. For parenting distress, mothers randomized to the intervention had a non-significant difference of −0.62 (95% CI: −4.90, 3.65) to control, after adjustment for baseline. Among fathers in the intervention, there was a non-significant difference of −1.28 (95% CI: −4.60, 2.04) compared to the control after adjustment for baseline. Differences in depressive symptoms followed a similar trend; no significant differences were found for either mothers or fathers. Among mothers randomized to the intervention, there was a non-significant difference of −0.57 (95% CI: −2.98, 1.84) compared to the control, after adjustment for baseline. As was found for mothers’ depressive symptoms scores, there was no significant difference between fathers in the intervention compared to those in the control after controlling for baseline (−0.91, 95% CI: −3.48, 1.67). 

At the family level, household chaos scores were similar at baseline and post-intervention. The difference of 0.65 (95% CI: −3.06, 1.77) was not statistically significant. 

### 3.4. 1-Year Follow-Up

Similar to the results at post-intervention, no intervention effect was observed for any of the stress measures among mothers or fathers at 1-year follow-up after controlling for baseline (Table 2). Specifically for general stress, the difference between the intervention and control was not significant (−0.15, 95% CI: −1.13, 0.83) after controlling for baseline. Among fathers, there was a non-significant difference in general stress at 1-year post-intervention after controlling for baseline (−0.90, 95% CI: −2.08, 0.27). The mean parental distress score at 1-year follow-up among mothers randomized to the intervention compared to the control yielded a non-significant difference of −1.92 (95% CI: −5.37, 1.53). Likewise, for fathers, the mean parental distress among those randomized to the intervention was not significantly different from the control at 1-year after controlling for baseline (−0.41, 95% CI: −4.56, 3.74). Among mothers randomized to the intervention, the mean depressive symptoms score was not significantly different from mothers randomized to the control (−0.92, 95% CI: −2.87, 1.04) after controlling for baseline. Among fathers, comparison of mean depressive symptoms scores at 1-year follow-up for the intervention and control resulted in a non-significant difference of −0.70 (95% CI: −2.98, 1.58) after controlling for baseline. 

At 1-year follow-up, mean household chaos among families randomized to the intervention compared to the control resulted in a non-significant difference of −2.57 (95% CI: −5.34, 0.21) after controlling for baseline. 

## 4. Discussion

The purpose of this study was to investigate differences in family-based stress between intervention and control groups at post-intervention and 1-year follow-up in a sample of Canadian mothers and fathers participating in the GFHS, a home-based obesity prevention randomized control trial. Our results suggest no harmful impact of the intervention program on the family environment across the four dimensions examined. 

The GFHS pilot studies demonstrated success in increasing children’s fruit and vegetable consumption [36] and at post-intervention, children and parents had lower indices of body fat [27,37]. This suggests that the GFHS intervention program did meaningfully change some family behaviours, but until the present study, it was unknown how these changes could impact families’ stress levels. 

The program may have encouraged families to implement more structured, organized behavioural patterns focused around these health goals, thus calming the home environment and increasing parenting confidence; however it is also possible that the program may have caused conflict or confusion from the disruptions to the families’ typical behaviours. Our results suggest that family stress levels were not different when comparing intervention to control families, despite evidence that behavioural changes did indeed occur among both parents and children [27,36,37]. 

There are several potential explanations for these results. Careful planning and consideration went into designing the GFHS intervention to have a minimal burden on participants, such as the health educator visits occurring within the family’s home instead of at a research centre, the use of online surveys to allow for more convenient completion, and financial compensation for the family’s time. Thus, participation in the study may not have been particularly burdensome to families. In addition, the use of motivational interviewing, a client-centred counselling technique that empowers participants to choose their own goals and strategies to achieve them [38], may have helped to relieve the burden from the participants compared to other more expert-led intervention techniques. The exact characteristics of the intervention protocol that contributed to these effects would require further research to disentangle but likely all factors had an influence. 

The current body of evidence on household stress is based mostly on clinical populations such as children with behavioural problems, developmental delays, or chronic illness [39,40,41,42], or special interest family situations such as parents who are military servicemembers or incarcerated [43,44,45], including the few studies that have examined family stress over the course of an intervention program [46,47,48,49]. This study extends evidence in the literature by providing insight into the impact of a home-based health intervention on the family environment in a community-based non-clinical sample of families. Additionally, our inclusion of both mothers’ and fathers’ perceptions addresses a substantial gap in the literature [34]. The present study also includes follow-up beyond the post-intervention period to better understand the nature of these associations. 

Despite this study’s many strengths, there are some limitations that merit consideration. First, these analyses are based on a small cohort of families because the GFHS pilot was not designed as a fully powered study; thus, there is a risk that important effects were not identified. Second, with respect to the general stress measure, a single item may not be sufficient to capture the many dimensions of everyday stress. Third, our protocol is to ask only Parent 1 (defined as the first parent to enrol in the study) items relating to the household; as such, it is possible that perceptions of the home environment chaos may differ between cohabitants. Fourth, the majority of families in our sample identified as Caucasian and nearly half had an annual household income of over $100,000, which limits the generalizability of our results. Additional research with a diverse sample of families is needed because the socio-cultural environment, including ethnic and economic factors, is an important consideration for parenting practices and family stress [50,51,52]. Finally, while it is most likely that any differences in stress due to the intervention would be evident in the post-intervention period, it is possible that the true nature of these associations requires a longer follow-up period to be discovered. Continued longer-term monitoring of the participants’ stress may be an important consideration for our participants’ retention in the study. Indeed, any family-focused or home-based intervention program should consider how disruptions to the family dynamic may influence participants’ willingness to adhere to the program. 

## 5. Conclusions

The GFHS has several behaviour change goals aimed at preventing childhood obesity; however, reducing family stress levels was not among the primary intentions of the program. While these results show no differences in family stress between the intervention and control groups, the overall mean stress levels seen here indicate that families may benefit from intervention strategies specifically aimed at reducing family stress. Program designs that integrate family physical and mental health promotion should be further investigated. In conclusion, these results demonstrate a need for continued research into how home-based health interventions influence the family environment. In particular, there is a need for intervention programs that incorporate specific stress-reduction messaging into family health programs.

## Figures and Tables

**Figure 1 nutrients-12-01835-f001:**
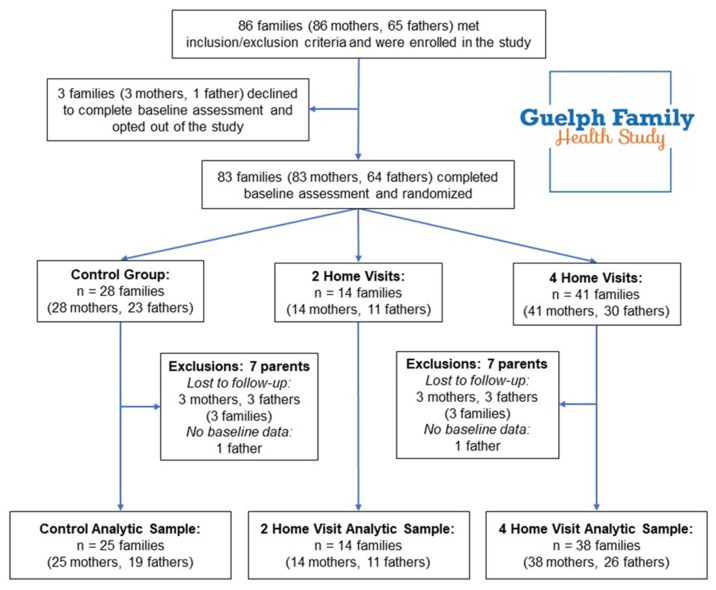
Study design and participant flow of the analytic sample from the Guelph Family Health Study Pilot Phase 1 and Phase 2 parent participants.

**Table 1 nutrients-12-01835-t001:** Baseline characteristics of parent participants in the Guelph Family Health Study.

**Characteristic (Individual)**	**Overall *n* = 133 Parents**	**Control *n* = 44 Parents**	**Intervention *n* = 89 Parents**
Baseline age (years), mean (SD)	35.5 (4.6)	34.8 (4.8)	35.9 (4.6)
Relation to child, n (%)			
Father	56 (42.1%)	19 (43.2%)	37 (41.6%)
Mother	77 (57.9%)	25 (56.8%)	52 (58.4%)
Ethnicity, n (%)			
White	109 (82.0%)	37 (84.1%)	72 (80.9%)
Other (e.g., Chinese, Latin American, South Asian, West Asian)	22 (16.5%)	5 (11.4%)	16 (18.9%)
Not reported	2 (1.5%)	2 (4.5%)	0 (0.0%)
Education, n (%)			
College diploma or less	30 (22.6%)	7 (15.9%)	23 (25.8%)
Some university or degree	35 (33.8%)	14 (38.1%)	31 (34.8%)
Postgraduate training	56 (42.1%)	21 (47.7%)	35 (39.3%)
Did not disclose	2 (1.5%)	2 (4.5%)	0 (0.0%)
**Characteristic (Family Level)**	***n* = 77 families**	***n* = 25 families**	***n* = 52 families**
Marital status, n (%)			
Married	66 (85.7%)	22 (88.0%)	44 (84.6%)
Other (i.e., living with partner, divorced)	11 (14.3%)	3 (12.0%)	8 (15.4%)
Annual household income, n (%)			
< $60,000	16 (20.8%)	5 (20.0%)	11 (21.2%)
$60,000 to $99,999	24 (31.2%)	5 (20.0%)	19 (36.5%)
$100,000+	34 (44.2%)	14 (56.0%)	20 (38.5%)
Not reported	3 (3.9%)	1 (4.0%)	2 (3.8%)
Number of children, n (%)			
1	17 (22.1%)	7 (28.0%)	10 (19.2%)
2	45 (58.4%)	15 (60.0%)	30 (57.7%)
3 or more	15 (19.5%)	3 (12.0%)	12 (23.1%)

**Table 2 nutrients-12-01835-t002:** Linear regression results comparing intervention and control groups with respect to stress levels at post-intervention and at 1-year follow-up after controlling for baseline, stratified by parent gender. Household chaos model analysed at the family level (one observation per household).

Measure	Intervention Group	Baseline Mean (SD)	Post-Intervention Mean (SD)	Difference from Control ^1^ β (95% CI) *p* value	1-Year Follow-Up Mean (SD)	Difference from Control ^1^ β (95% CI) *p* Value
**Analysis of mothers in the home visit groups (n = 52) compared to the control group (n = 25)**						
**General Stress**	Intervention	6.60 (2.02)	6.02 (2.10)	−0.60 (−1.47, 0.27) 0.18	6.55 (1.89)	−0.15 (−1.13, 0.83) 0.76
	Control	6.32 (2.12)	6.63 (1.69)		6.63 (1.92)	
**Parenting Distress**	Intervention	28.04 (9.70)	26.80 (8.91)	−0.62 (−4.90, 3.65) 0.77	28.49 (8.91)	−1.92 (−5.37, 1.53) 0.27
	Control	29.68 (6.64)	28.13 (9.28)		30.91 (8.26)	
**Depressive Symptoms**	Intervention	6.78 (5.39)	5.98 (5.23)	−0.57 (−2.98, 1.84) 0.64	6.00 (4.48)	−0.92 (−2.87, 1.04) 0.35
	Control	6.80 (4.34)	6.67 (5.01)		6.73 (4.73)	
**Analysis of fathers in the home visit groups (n = 37) compared to the control group (n = 19)**						
**General Stress**	Intervention	6.78 (1.86)	6.72 (1.63)	0.56 (−0.43, 1.56) 0.26	6.03 (2.14)	−0.90 (−2.08, 0.27) 0.13
	Control	6.26 (2.10)	5.88 (2.20)		6.57 (2.06)	
**Parenting Distress**	Intervention	29.03 (8.03)	27.81 (7.16)	−1.28 (−4.60, 2.04) 0.44	28.03 (8.12)	−0.41 (−4.56, 3.74) 0.84
	Control	27.53 (5.50)	28.65 (5.72)		27.57 (4.38)	
**Depressive Symptoms**	Intervention	7.19 (5.25)	6.13 (4.63)	−0.91 (−3.48, 1.67) 0.48	6.57 (4.31)	−0.70 (−2.98, 1.58) 0.54
	Control	7.63 (3.39)	7.06 (4.38)		7.86 (3.23)	
**Analysis of families in the home visit groups (n = 52) compared to control group (n = 25)**						
**Household Chaos**	Intervention	31.02 (8.39)	30.92 (8.07)	0.65 (−3.06, 1.77) 0.60	30.29 (7.89)	−2.57 (−5.34, 0.21) 0.07
	Control	31.04 (6.31)	31.74 (6.45)		33.00 (6.20)	

^1^ Linear regression coefficient after controlling for baseline.

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
