# Peer review of "Parent Stress as a Consideration in Childhood Obesity Prevention: Results from the Guelph Family Health Study, a Pilot Randomized Controlled Trial"

_nutrients, 2020, doi:10.3390/nu12061835_

Round 1
Reviewer 1 Report
The purpose of this study was to 1) examine the longitudinal impact of participation in a family-based health promotion intervention on parent perceived general life stress, parent distress, depressive symptoms, and household chaos relative to an attention-control condition, and 2) whether parent gender moderated the impact of the intervention on stress-related outcomes. The authors conclude that there were no significant differences in stress between intervention and control groups at post-intervention or 1-year follow-up when stratified by gender.
The authors did a good job of illustrating the importance of examining the impact of a family-based health promotion program on parental stress outcomes, given the detrimental impact of parent stress on child health outcomes and intervention effectiveness. The findings of this study have the potential to provide a unique contribution to the field. I have a few suggestions/queries for the authors.
- A household’s income level could affect the impact of a family-based health promotion intervention on parental stress as those with more financial resources may also have greater access to child care resources, resources to care for aging parents, healthy foods, physical activity facilities, etc. Was household income level controlled for in analyses?
- Were any demographic variables controlled for in analyses? If not, I am assuming it is due to small sample size, but it would be helpful to have clarification in the statistical methods section since education level, number of children, marital status, and ethnicity could all seemingly affect stress levels.
- Why are mean stress levels overall (i.e., not stratified by gender) not reported?
- It would be informative to see the initial linear regression models that showed significant gender X stress interaction terms, especially since the post hoc analyses do not show any significant impact of the intervention on stress by gender. Are there main effects of the intervention on stress?
- Table 2 - some of the means and SDs are not stacked, which disrupts the uniformity of the table (e.g., the means and SDs for mother general stress and baseline)
- Sections 3.3 Post-Intervention and 3.3. 1-Year Follow-Up do not need to include the means, standard deviations, and confidence intervals that are already listed in Table 2. This is redundant and makes the narrative more challenging to read.
- According to Table 2, there is a statistically significant difference in household chaos between intervention and control families at 1-year follow-up (p=0.07), but this is not reported in section 3.3. 1-Year Follow-Up. Please clarify.
- Among the intervention group, how many household visits were completed? The number of home visits the health educator was able to conduct with families could impact stress. Intervention dose could be a significant variable to control for in analyses.
Author Response
Thank you to the reviewers for their helpful and thorough review of our manuscript. Our responses are listed below each comment, and any modifications made to the manuscript are shown in red and underlined in the main document.
Reviewer 1:
The purpose of this study was to 1) examine the longitudinal impact of participation in a family-based health promotion intervention on parent perceived general life stress, parent distress, depressive symptoms, and household chaos relative to an attention-control condition, and 2) whether parent gender moderated the impact of the intervention on stress-related outcomes. The authors conclude that there were no significant differences in stress between intervention and control groups at post-intervention or 1-year follow-up when stratified by gender.
The authors did a good job of illustrating the importance of examining the impact of a family-based health promotion program on parental stress outcomes, given the detrimental impact of parent stress on child health outcomes and intervention effectiveness. The findings of this study have the potential to provide a unique contribution to the field. I have a few suggestions/queries for the authors.
Response: Thank you for your support of our research.
1.1 A household’s income level could affect the impact of a family-based health promotion intervention on parental stress as those with more financial resources may also have greater access to child care resources, resources to care for aging parents, healthy foods, physical activity facilities, etc. Was household income level controlled for in analyses?
Response: We did not include household income as a covariate because this was a randomized trial and income was not used as a blocking variable. Had this been an observational study, investigation of potential confounders would have been in order. Note that initial descriptive statistics did not identify systematic differences among intervention groups with respect to income. For example, the mean income was very similar between the treatment and control groups (summarized below).
|
Study Intervention Group |
Number of Families |
Mean Annual Household Income |
Standard Deviation |
|
2 Home Visits |
14 |
89642.86 |
35217.14 |
|
4 Home Visits |
36 |
94583.33 |
40273.62 |
|
Control |
24 |
103750.00 |
37161.52 |
1.2 Were any demographic variables controlled for in analyses? If not, I am assuming it is due to small sample size, but it would be helpful to have clarification in the statistical methods section since education level, number of children, marital status, and ethnicity could all seemingly affect stress levels.
Response: Indeed, these factors are important considerations for family stress. As with our response to comment 1.1 above, this was a randomized trial and none of the suggested variables were included in the design (i.e., none were included as blocking variables). Our methods section (page 6, lines 219-221) has been updated to clarify this, as follows:
“No demographic covariates were included in the model. The use of a randomized design would mean that any difference in demographic characteristics across study groups would be due to chance.”
1.3 Why are mean stress levels overall (i.e., not stratified by gender) not reported?
Response: Fathers have been historically excluded from family health literature and so comparisons to other work is limited without stratification. Unstratified means and standard deviations for the stress measures stratified in the manuscript are presented in the table below; they are very similar to the stratified results. Please note that the household chaos measure was analysed at the family-level and was not stratified in the main manuscript.
|
Stress Measure |
Timepoint |
Intervention Mean (SD) |
Control Mean (SD) |
|
General Stress |
Baseline |
6.68 (1.94) |
6.30 (2.09) |
|
Post-Intervention |
6.29 (1.95) |
6.32 (1.93) |
|
|
1-Year Follow-Up |
6.35 (1.99) |
6.61 (1.95) |
|
|
Parenting Distress |
Baseline |
28.45 (9.00) |
28.75 (6.20) |
|
Post-Intervention |
27.20 (8.24) |
28.34 (7.92) |
|
|
1-Year Follow-Up |
28.32 (7.62) |
29.61 (7.13) |
|
|
Depressive Symptoms |
Baseline |
6.95 (5.30) |
7.15 (3.94) |
|
Post-Intervention |
6.04 (4.98) |
6.83 (4.71) |
|
|
1-Year Follow-Up |
6.22 (4.40) |
7.17 (4.20) |
1.4 It would be informative to see the initial linear regression models that showed significant gender X stress interaction terms, especially since the post hoc analyses do not show any significant impact of the intervention on stress by gender. Are there main effects of the intervention on stress?
Response: Upon closer inspection, the rationale for stratified analyses was misinterpreted and misstated. We apologize for the confusion. There was never an interaction between gender and intervention in any models. Interactions between baseline stress and gender are not of inherent interest but the presence of this type of interaction in two models provided additional rationale for the stratified models. Because of the widely-reported gender differential in stress research compounded with the exclusion of fathers from family health research, we decided to provide stratified results only. To present unstratified results would complicate comparison to other literature based on mothers only, as well as downplay the fathers’ contributions to the research. We have corrected the text regarding significant interaction terms and added additional support for the decision to stratify in the manuscript (page 6, lines 216-218). For your information, the main effects of the intervention on stress (i.e., not stratified by gender) are presented below.
|
Stress Measure |
Timepoint |
β (95% CI) P value |
|
General Stress |
Post-Intervention |
-0.13 (-0.77, 0.51) 0.70 |
|
1-Year Follow-Up |
-0.39 (-1.15, 0.37) 0.31 |
|
|
Parenting Distress |
Post-Intervention |
-0.82 (-3.83, 2.19) 0.60 |
|
1-Year Follow-Up |
-1.26 (-3.82, 1.30) 0.33 |
|
|
Depressive Symptoms |
Post-Intervention |
-0.70 (-2.43, 1.03) 0.43 |
|
1-Year Follow-Up |
-0.83 (-2.27, 1.03) 0.43 |
1.5 Table 2 - some of the means and SDs are not stacked, which disrupts the uniformity of the table (e.g., the means and SDs for mother general stress and baseline)
Response: This formatting change has been made; all means and SDs are now stacked.
1.6 Sections 3.3 Post-Intervention and 3.3. 1-Year Follow-Up do not need to include the means, standard deviations, and confidence intervals that are already listed in Table 2. This is redundant and makes the narrative more challenging to read.
Response: We have edited the results writeup to reduce this redundancy (pages 6-8, lines 247-307).
1.7 According to Table 2, there is a statistically significant difference in household chaos between intervention and control families at 1-year follow-up (p=0.07), but this is not reported in section 3.3. 1-Year Follow-Up. Please clarify.
Response: While this result does approach the commonly accepted threshold for significance (p < 0.05), the 95% confidence interval for this result does contain zero, indicating the potential that the true difference between groups may be nonexistent. As such, we are hesitant to identify this result as significant.
1.8 Among the intervention group, how many household visits were completed? The number of home visits the health educator was able to conduct with families could impact stress. Intervention dose could be a significant variable to control for in analyses.
Response: With the exception of the families lost to follow-up, all families in the intervention groups received 100% of the home visits. We have included this in our Exclusions and Losses to Follow Up section (page 3, lines 131-134), as follows:
“The remaining 82 families (146 parents) completed all components of the intervention program, though 5 families (5 mothers, 6 fathers) were later lost to follow-up resulting in a 92.8% retention rate of the GFHS Pilot 1 and 2 cohorts at 1-year post-intervention.”
Reviewer 2 Report
I thought the paper was well written and aligned with the main aims of the Journal.
The study aims to examine the impacts of a family-based intervention on reported stress levels post program and at follow up. I appreciated the analysis broken up by gender - mother versus father - as the impact of stress is differential based on gender role. However, I would have liked to see the analysis by child age since the stress can be impacted by the role of the parent and how that differs by child age.
The use of the control group is helpful to control for external stressors. The methods were appropriate given the state of the literature and the findings are promising in an area of research that is incredibly understudied.
The authors identify the limitation of the primarily Caucasian affluent sample which is helpful but they could add some indication of research that shows stress among different ethnic groups varies especially when it comes to the parenting literature.
Author Response
Thank you to the reviewers for their helpful and thorough review of our manuscript. Our responses are listed below each comment, and any modifications made to the manuscript are shown in red and underlined in the main document.
Reviewer 2:
I thought the paper was well written and aligned with the main aims of the Journal.
Response: Thank you for your support of our research.
2.1 The study aims to examine the impacts of a family-based intervention on reported stress levels post program and at follow up. I appreciated the analysis broken up by gender - mother versus father - as the impact of stress is differential based on gender role. However, I would have liked to see the analysis by child age since the stress can be impacted by the role of the parent and how that differs by child age.
Response: Child age was not included as a covariate in these analyses because the majority of parents in this sample (78%) have more than one child and it would be difficult to statistically account for multiple child ages. We expect that the relatively narrow age bracket of this cohort (18-months to 5-years at baseline) for both the intervention and control groups helps to mitigate the potential impact of child age on parent stress.
2.2 The use of the control group is helpful to control for external stressors. The methods were appropriate given the state of the literature and the findings are promising in an area of research that is incredibly understudied.
Response: Thank you for your review.
2.3 The authors identify the limitation of the primarily Caucasian affluent sample which is helpful but they could add some indication of research that shows stress among different ethnic groups varies especially when it comes to the parenting literature.
Response: We have included the following sentence with the references listed below to address this limitation (page 10, lines 362-363).
“Additional research with a diverse sample of families is needed because the socio-cultural environment, including ethnic and economic factors, is an important consideration for parenting practices and family stress.”
Harkness, S., & Super, C. (1995). Culture and Parenting. The Handbook of Parenting, 2(Biology and ecology of parenting), 211–234. papers2://publication/uuid/10FF018B-3E0A-4C95-9DA5-A51BE96AF446
Hoff, E., Laursen, B., & Tardiff, T. (2002). Socioeconomic Status and Parenting. In Handbook of Parenting (pp. 231–252). Lawrence Erlbaum Associates Publishers.
Nomaguchi, K., & House, A. N. (2013). Racial-Ethnic Disparities in Maternal Parenting Stress. Journal of Health and Social Behavior, 54(3), 386–404. https://doi.org/10.1177/0022146513498511
Reviewer 3 Report
Dear Authors,
This study explored how family stress levels were impacted by participation in a home-based obesity prevention intervention in Canada.
This is valuable research in light of childhood overweight and obesity. Results indicate no significant differences between intervention 24 and control groups for any stress measure at any time point, indicating a neutral effect of the GFHS 25 intervention on family stress.
Abstract
It is complete and well structured. It provides the article’s main information about background, methods, results and conclusion.
Introduction
The introduction is exhaustive and provides a complete overview of the matter.
The first paragraph mentions the questions and issues that outline the background of the study and establishes, the context and relevance of the problem.
The main part includes the importance of the problem and unclear issues.
The last paragraph states the main objective and the questions addressed in the manuscrit.
The only suggestion to be made would be to try to reduce The introduction to 1 page (in 3 paragraphs).
Materials and methods
Methods are adequately described. This information provides to the journal audience a better and complete understanding of the manuscript.
It should be noted the that sample, 77 families, is not very large, but it is acceptable due to its peculiarity. Data could be a limitation, as it is not recent (between 2014 and 2016).
Results
This section includes the necessary data of the experimental results. All tables are clear and have a short explanatory title and caption.
Discussion and Conclusions
Although this is not a mandatory section, the autors have included it. It provides a clear idea of the main results of the investigation and its limitations.
I am looking forward to reading the final version of the manuscript.
Author Response
Thank you to the reviewers for their helpful and thorough review of our manuscript. Our responses are listed below each comment, and any modifications made to the manuscript are shown in red and underlined in the main document.
Reviewer 3:
Dear Authors,
This study explored how family stress levels were impacted by participation in a home-based obesity prevention intervention in Canada.
This is valuable research in light of childhood overweight and obesity. Results indicate no significant differences between intervention 24 and control groups for any stress measure at any time point, indicating a neutral effect of the GFHS 25 intervention on family stress.
Response: Thank you for your support of our research.
3.1 Abstract: It is complete and well structured. It provides the article’s main information about background, methods, results and conclusion.
Response: Thank you for your review.
3.2 Introduction: The introduction is exhaustive and provides a complete overview of the matter. The first paragraph mentions the questions and issues that outline the background of the study and establishes, the context and relevance of the problem. The main part includes the importance of the problem and unclear issues. The last paragraph states the main objective and the questions addressed in the manuscript. The only suggestion to be made would be to try to reduce the introduction to 1 page (in 3 paragraphs).
Response: We have condensed the introduction to a single page (pages 1-3, lines 30-108).
3.3 Materials and methods: Methods are adequately described. This information provides to the journal audience a better and complete understanding of the manuscript. It should be noted the that sample, 77 families, is not very large, but it is acceptable due to its peculiarity. Data could be a limitation, as it is not recent (between 2014 and 2016).
Response: Thank you for your review of our methodology. We have clarified our sentence on the time period to better reflect the entire data collection period, not just baseline data collection (page 5, lines 169-170), as follows:
“Baseline data were collected between December 2014 and November 2016 at the University of Guelph, Ontario, Canada; follow-up data collection was completed by November 2018.”
3.4 Results: This section includes the necessary data of the experimental results. All tables are clear and have a short explanatory title and caption.
Response: Thank you for your review.
3.5 Discussion and Conclusions: Although this is not a mandatory section, the authors have included it. It provides a clear idea of the main results of the investigation and its limitations. I am looking forward to reading the final version of the manuscript.
Response: Thank you for your review.
Round 2
Reviewer 1 Report
The authors did a nice job of clarifying their methodology and the purpose of the study - which is to investigate whether participation in a family-based health intervention differentially impacts parent perceptions of stress based on gender of the parent.